# Positive Association between Dietary Inflammatory Index and the Risk of Osteoporosis: Results from the KoGES_Health Examinee (HEXA) Cohort Study

**DOI:** 10.3390/nu10121999

**Published:** 2018-12-17

**Authors:** Hye Sun Kim, Cheongmin Sohn, Minji Kwon, Woori Na, Nitin Shivappa, James R. Hébert, Mi Kyung Kim

**Affiliations:** 1Cancer Epidemiology Branch, Division of Cancer Epidemiology and Prevention, National Cancer Center, Goyang, Gyeonggi do 10408, Korea; hyeskim10@gmail.com (H.S.K.); 74433@ncc.re.kr (M.K.); 2Department of Food and Nutrition, Wonkwang University, Iksan 54538, Korea; ccha@wku.ac.kr (C.S.); nawoori6@gmail.com (W.N.); 3Cancer Prevention and Control Program, University of South Carolina, Columbia, SC 29208, USA; SHIVAPPA@mailbox.sc.edu (N.S.); JHEBERT@mailbox.sc.edu (J.R.H.); 4Department of Epidemiology and Biostatistics, Arnold School of Public Health, University of South Carolina, Columbia, SC 29208, USA; 5Connecting Health Innovations LLC, Columbia, SC 29201, USA

**Keywords:** osteoporosis, inflammation, nutrition, epidemiology

## Abstract

Previous studies have found that diet’s inflammatory potential is related to various diseases. However, little is known about its relationship with osteoporosis. The aim of this study was to investigate the association between the dietary inflammatory index (DII^®^) and osteoporosis risk in a large-scale prospective cohort study in Korea. This prospective cohort study included 159,846 participants (men 57,740; women 102,106) from South Korea with a mean follow-up of 7.9 years. The DII was calculated through a validated semi-quantitative FFQ (SQFFQ), and information on osteoporosis was self-reported by the participants. Analyses were performed by using a multivariable Cox proportional hazard model. Higher DII scores were associated with higher osteoporosis risk (HR 1.33; 95% CI 1.12–1.58). In women, a higher DII score indicated a higher risk of osteoporosis (HR 1.33; 95% CI 1.11–1.59). However, a hazards ratio of similar magnitude in men was not significant (HR 1.32; 95% CI 0.64–2.71). Post-menopausal women had higher risks of osteoporosis for higher DII scores (HR 1.33; 95% CI 1.09–1.63), whereas among pre-menopausal women, the relationship was not statistically significant (HR 1.39; 95% CI 0.87–2.21). Also, there was an increase in osteoporosis risk when the DII increased among women participants with irregular physical activity (HR 1.53; 95% CI 1.17–2.01); however, there was no statistically significant increase in osteoporosis risk among women participants with regular physical activity (HR 1.19; 95% CI 0.93–1.52). A more pro-inflammatory diet was significantly associated with higher osteoporosis risk in women. Given the similar magnitude of the hazards ratio, studies with sufficient numbers of men are warranted.

## 1. Introduction

Osteoporosis is an age-associated disease that is greatly influenced by genetic, epigenetic, and environmental factors [1]. It is characterized by a decreased density of normally mineralized bone, which leads to a decreased mechanical strength that makes individuals susceptible to fractures, pain, and disability [2]. As increasing age is closely related to increasing risk of osteoporosis, it has gained attention as an important public threat that needs to be addressed as populations grow older. Not only does osteoporosis threaten the health of a population, it also takes huge personal and economic tolls [3]. In Korea, the prevalence of osteoporosis in the last decade increased from 6.1% in men and 24.3% in women, to 13.1% in men and 35.5% in women, respectively [4]. Thus, the development of strategies for the prevention of osteoporosis is important, in order to lessen the pain, suffering, and economic burdens posed by this disease.

Chronic, systemic inflammation increases the risk of various ailments, such as cardiovascular and periodontal diseases [5,6,7]. Several researchers have found that chronic inflammation is not only a potential risk factor for osteoporosis, but is also linked with factors that are critical for bone physiology [8]. A variety of cells, including macrophages and neutrophils, secrete interleukin (IL)-1, IL-6, and tumor necrosis factor- α (TNF-α), which are pro-inflammatory cytokines that are important inflammatory mediators. IL-1 and TNF-α work as major triggers for osteoclast activation, and IL-6 collaborates with other bone-resorbing agents [9,10,11]. Additionally, Ganesan et al. observed a negative association between high-sensitivity C-reactive protein (hs-CRP) produced by IL-6, and bone-mineral density (BMD) [12], on basis of which they suggested a relationship between inflammation and osteoporosis [13,14]. Koh et al., meanwhile, also found a negative association between hs-CRP levels and BMD [12]. These results all contribute to growing evidence for an association between inflammation and osteoporosis.

Various studies have found evidence indicating that an individual’s habitual diet influences inflammation. Researchers who conducted studies on the impact of diet on inflammation inferred that diet influences an individual’s inflammatory response by incurring exposure that accumulates in the body [15,16]. The dietary inflammatory index (DII^®^), which is a literature review-based score measuring a diet’s inflammatory potential, has been validated against inflammatory markers such as CRP and IL-6 [17,18]. Several studies have found that patients with a high DII score showed more significant increases in hs-CRP levels than did patients with a low DII score, which suggests the possible utilization of the DII for determination of the association between diet’s inflammatory potential and chronic diseases [19,20]. Several epidemiological studies have been conducted to examine the associations between the DII and various ailments such as fracture, cardiovascular diseases (CVD), and cancer [7,21,22,23,24]. Osteoporosis is the main risk factor for fracture, and osteoporosis and fracture are closely related in clinical settings; nevertheless, osteoporosis is a potential target for the efficient prevention of fracture. However, to the best of our knowledge, no studies on the possible associations between the DII and osteoporosis have yet been reported. It is unclear as to whether a more pro-inflammatory diet is associated with osteoporosis. Knowledge of the association of inflammatory diet with osteoporosis risk could be important for the tailoring of intervention strategies that are related to diet modulation for reduced inflammation. In light of the potential usefulness of such associations in efforts to develop strategies to prevent of osteoporosis, the aim of the present study was to examine, by means of a large-scale prospective cohort study, the associations between the inflammatory potential of diet, as measured by the DII and osteoporosis risk. The study’s hypothesis was that participants with higher DII scores, indicative of more pro-inflammatory diets, are at higher risk of developing osteoporosis.

## 2. Materials and Methods

### 2.1. Study Population and Data Collection

The present study used data from the Korean Genome and Epidemiology Study (KoGES), a cohort study based on the general Korean population, to gain information on the genetic, environmental and lifestyle determinants of osteoporosis. Several cohorts in the KoGES (i.e., KoGES_Ansan and Ansung study, the KoGES_cardiovascular disease association study (CAVAS), and the KoGES_health examinee study (HEXA)) included subjects who had been recruited from the National Health Examinee Registry, and all of whom were 40–79 years of age at the baseline. Among those cohorts, data from the KoGES-HEXA study was used in the present study to determine the association between the DII and osteoporosis risk. Detailed information on KoGES can be found elsewhere [25].

To summarize, participants were asked to voluntarily fill out a baseline survey through letters, campaigns, on-site invitations, and conferences in the community. A total of 173,343 participants (59,291 men, 114,052 women) gathered from 38 health examination centers and hospitals in Korea and were asked to attend follow-up visits held between 2007 and 2016 (mean: 7.9 years). Data from the baseline survey to the first follow-up were used in the present study, and person-years of exposure time were accumulated from the baseline measure until the first follow-up date. Among the original 173,343 participants, those who had osteoporosis at the baseline or who had missing data (=9989) were excluded. Those with unreliable caloric intakes (men <500 Kcal or >6000 Kcal; women <500 Kcal or >4000 Kcal = 1208) or had no energy data were also excluded, which left 159,846 (57,740 men, 102,106 women) remaining subjects. The outcome was defined as a diagnosis of osteoporosis by a medical doctor, which was self-reported by the participants. A total of 2572 incident osteoporosis cases (148 men, 2424 women) were identified after the exclusion of observations with missing covariates (Figure 1). Informed consent was obtained from all of the study participants prior to the study. The Institutional Review Board of the National Cancer Center in Korea approved the statistical analyses and all of the methods used in the present study; also, all relevant guidelines and regulations were followed (IRB No. NCC2018-0164).

### 2.2. Dietary Assessment Using SQ-FFQ and the Calculation of DII

A validated semi-quantitative food frequency questionnaire (SQFFQ) was used to measure dietary food intakes at baseline. A test of the validity of the SQFFQ was conducted in a previous study [26]. The study subjects estimated their consumption frequencies and the average amounts of 106 food items consumed over the course of one year. The total of the values of the average serving amounts, portions per unit, and serving frequencies, was applied to the measurement of the nutrient intake per day [27]. The frequencies of each food item consumed during the past year comprised nine choices, starting from “almost never” and ending with “more than three times per day.” Three portion-size answers were available: 1/2 standard serving, one standard serving, and 3/2 serving. Total energy and nutrient intakes were calculated using a food composition table [28]. 

Details on the DII are available elsewhere [29]. In brief, a low DII score represents an anti-inflammatory diet, while a high DII score represents a pro-inflammatory diet. Pro-inflammatory parameters include total calories, protein, cholesterol, carbohydrate, total fat, saturated fatty acids, and vitamin B-12. Anti-inflammatory parameters include niacin, riboflavin, magnesium, vitamin C, vitamin E, beta-carotene, flavan-3-ol, flavonones, isoflavones, fiber, ginger, onion, MUFAs, PUFAs, thiamin, vitamin B-6, vitamin A, vitamin D, folic acid, anthocyanidins, flavonols, flavones, alcohol, garlic, pepper, and tea [29]. The nutritional content data used in the present study were excerpted from the Functional Ingredients Table (Rural Development Administration), Computer Aided Nutritional Analysis (The Korean Nutrition Society), and data were provided by the U.S. Department of Agriculture. 

In this study, 37 of the possible 45 food parameters were scored according to their effects on the levels of inflammatory markers, including IL-1β, IL-6, CRP, IL-10, IL-4, and TNF-α. In this updated version of the DII, 1943 articles were reviewed and scored. Forty-five food parameters, including foods, nutrients, and other bioactive compounds, were identified based on their inflammatory effect on six specific inflammatory markers, including CRP, IL-1β, IL-4, IL-6, IL-10, and tumor necrosis factor (TNF)-α. A regionally representative world database representing diet surveys from 11 countries was used as a comparative standard for each of the 45 parameters (i.e., foods, nutrients, and other food components). Intake values from this database were used to calculate the DII scores. This is explained in more detail in the DII Methods paper [29]. Briefly, a standard mean for each parameter from the representative world database was subtracted from the actual individual exposure and divided by its standard deviation to generate Z scores. These Z scores were converted to proportions (thus minimizing effects of outliers/right-skewing). These values were then doubled, and 1 was subtracted, to achieve symmetrical distribution with values centered on ≈0. The resulting value was then multiplied by the corresponding inflammatory score for each food parameter, and summed across all food parameters, to obtain the overall DII score. The associations between inflammatory markers and the DII have been validated in several studies that are based on different populations [29,30,31,32]. In brief, high sensitivity CRP measurements were used to construct the validity of the DII score in a longitudinal cohort, using multiples (up to 15) of 24-hr dietary recall interviews, and up to five 7-day dietary recalls. The DII was subsequently validated in four studies among different populations with a variety of inflammatory biomarkers (i.e., interleukin (IL)-6, hs-CRP, fibrinogen, homocysteine, and TNF-α) [20,23,30,31,32,33]. Both intakes from foods and supplements were included in the DII calculation.

### 2.3. Covariates

During the baseline and follow-up examinations, participants completed a questionnaire that included questions on sociodemographics, personal and family medical history, regularity of physical activity, and the SQFFQ, which has been described elsewhere [26]. Age, calcium intake, and energy intake were measured as continuous variables, and educational attainment was categorized in three ways: elementary school or below, middle school to high school, and college or above. Also, menopausal status was divided into pre- or peri-menopause (currently experienced monthly menstrual cycle) and post-menopause (one year or more without menstrual cycle). As for smoking status, subjects who had smoked more than 400 cigarettes to date and continued to smoke at the time of the survey were sorted as “current” smokers; those who had never smoked more than 400 cigarettes were sorted as “never” smokers, and those who had smoked approximately 400 cigarettes but were refraining from smoking at the time of the survey, as “past” smokers. With regard to alcohol consumption, those who had consumed alcohol and still considered themselves to be drinkers at the time of the survey were categorized as “current” drinkers; those who had never consumed alcohol were classified as “never” drinkers, and those who had consumed alcohol but were refraining from drinking at the time of the survey, as “past” drinkers. Regularity of physical activity was determined according to whether or not subjects participated regularly in any sports to the point of sweating. Those who did so were classified as the “regular” physical activity group, while those who did not were classified as the “irregular” physical activity group.

### 2.4. Statistical Analysis

In order to analyze the associations of the DII with the characteristics and risks of osteoporosis, the DII was divided into five levels (quintiles), based on the cohort for whom there was no baseline osteoporosis. Continuous variables were expressed as means with standard deviation, and categorical variables were expressed as frequency numbers with percentages. The Jonckheere–Terpstra test was used to measure the *p* values for trends in the continuous variables, while the Mantel–Haenszel Chi-square test was applied to the categorical variables. The multivariable Cox proportional hazard model was used to find any association of the baseline DII with newly diagnosed osteoporosis. To confirm the assumption of proportional risk, all of the models were evaluated and deemed to be consistent with a model that included time-dependent covariates. The multivariable-model was adjusted for sex (for all subjects); age (categorical); body mass index (BMI) (categorical); smoke (categorical); calcium intake (continuous); alcohol consumption (categorical); physical activity (categorical); energy intake (continuous). Additionally, multiple imputation with five complete datasets (which had been compiled by accounting for missing data in baseline covariates) was performed [34]. The HRs were calculated as 95% confidence intervals, and the two-sided probability values were <0.05 statistically significant. All of the statistical analyses were performed with SAS^®^ 9.3 (SAS Institute, Cary, NC, USA). *p* values < 0.05 were considered as statistically significant.

## 3. Results

Analyses are based on 59,291 male and 114,502 female cohort members with evaluable data. Men had a higher median DII score (0.929) than did women (0.877). The baseline characteristics of the 173,343 participants, based on the DII quintiles are presented in Table 1. The mean age of the participants increased, but calcium intake decreased as the DII increased (*p* < 0.0001), while the mean energy intake decreased as the DII increased. The number of women in the lower DII range was larger than that in the higher DII range (*p* < 0.0001). Also, the number of never smokers increased as the DII decreased, while the numbers of past smokers and current smokers increased as the DII increased (*p* < 0.0001). Additionally, participants with higher DII levels relative to those with lower levels had significantly lower educational attainment, as well as lower income, and lower BMI, and they exercised irregularly (*p* < 0.0001, respectively). The number of women participants who were in the post-menopausal state increased as the DII increased, and the number of people who were married increased as the DII decreased (*p* < 0.0001, respectively). Lastly, the number of never drinkers increased as the DII increased, while the number of current drinkers decreased.

During the 7.9-year follow-up, 2572 individuals (148 men and 2424 women; = 1.64% of population at baseline) developed osteoporosis. Cox’s regression analysis, according to which nine possible confounding variables at the baseline were adjusted set the lowest DII as the reference (=Quintile 1), and suggested that subjects with the highest DII score (=Q5) had a significantly higher risk of developing osteoporosis (HR 1.33; 95% CI 1.12–1.58; Table 2). After imputation, a similarly positive association between the DII and osteoporosis risk was observed (HR 1.32; 95% CI 1.12–1.57; Table 2).

Table 2 reports the results stratified by sex. As can be seen, similar results were observed: women who had higher DII scores (=Q5) had a significantly higher osteoporosis risk (HR 1.33; 95% CI 1.11–1.59; Table 2). Correspondingly too, a statistically significant positive association between the DII and osteoporosis risk was observed after imputation (HR 1.33; 95% CI 1.11–1.58; Table 2). Despite a nearly identical hazard ratio, no significant association was observed in men (HR 1.32; 95% CI 0.64–2.71; Table 2), and it did not reach statistical significance after imputation (HR 1.27; 95% CI 0.63–2.60; Table 2).

The associations between the DII and osteoporosis risks, stratified by menopausal status and physical regularity, also were determined. The results as stratified by menopausal status were significant: post-menopausal women had higher risks of osteoporosis for higher DII scores (HR 1.33; 95% CI 1.09–1.63; Table 3), while no such association was evident among pre- or peri-menopausal women, despite a hazards ratio of even greater magnitude (HR 1.39; 95% CI 0.87–2.21; Table 3). Similarly, after imputation, the association between the DII and osteoporosis risk was significant among post-menopausal women (HR 1.33; 95% CI 1.09–1.61; Table 3), and the association among pre- or peri-menopausal women did not achieve statistical significance after imputation (HR 1.37; 95% CI 0.87–2.18; Table 3). 

Osteoporosis risk increased significantly as the DII increased among subjects who did not participate regularly in any sports to the point of sweating (HR 1.49; 95% CI 1.14–1.93; Table 4); however, the association was attenuated, and it did not reach significance among people who had regular physical activity (HR 1.21; 95% CI 0.96–1.53; Table 4). After imputation, the association between the DII and osteoporosis risk remained significant among participants with irregular physical activity (HR 1.51; 95% CI 1.16–1.95; Table 4), while that among those with regular physical activity was not (HR 1.21; 95% CI 0.96–1.52; Table 4). Among women, those with irregular physical activity also had a significantly higher risk of osteoporosis (HR 1.53; 95% CI 1.17–2.01; Table 4), while those with regular physical activity showed non-significant results (HR 1.19; 95% CI 0.93–1.52; Table 4). After imputation, similar results were observed among those with irregular and regular physical activity (HR 1.56; 95%CI 1.20–2.04; HR 1.19; 95% CI 0.93–1.51, respectively; Table 4). By contrast, both men with irregular and regular physical activity showed non-significant results (HR 0.86 95% CI 0.30–2.53; HR 1.67; 95% CI 0.63–4.46, respectively; Table 4). Similarly, the non-significant results were observed after imputation among men with irregular and regular physical activity (HR 0.84; 95% CI 0.30–2.38; HR 1.65; 95% CI 0.62–4.40, respectively; Table 4).

## 4. Discussion

This prospective cohort study aimed to find the association between diet’s inflammatory potential, as indicated by the DII score, and the osteoporosis risk in a large-scale prospective cohort study. Our results confirmed the study’s hypothesis that higher DII scores, indicative of more pro-inflammatory diets, were associated with higher osteoporosis risk. During the 7.9-year follow-up, individuals that had the most pro-inflammatory diet (i.e., the highest DII score) had a 33% higher risk (32% after imputation) of getting osteoporosis, than those with the least pro-inflammatory diet (i.e., the lowest DII score), after adjusting for potential confounding variables at the baseline. These results suggested a positive relationship between inflammation and risk of osteoporosis, which can be caused by the secretion of cytokines from inflammatory cells that cause bone resorption. However, while women showed a significant association between DII score and osteoporosis risk, those with the highest DII score had a 33% higher risk of getting osteoporosis. Men showed a similar, 32%, increase in risk. However, it was not statistically significant, perhaps in large part because of the relatively smaller sample size of men (i.e., 34% of the cohort) versus women (i.e., 66% of the cohort).

Although there has been no study on any association between the DII and osteoporosis that can be used to confirm the results of the current study, to the best of our knowledge, several investigations into a possible association between the DII and fracture reported similar results. For example, one study that was conducted based on data from the Women’s Health Initiative Observational Study and clinical trials reported results showing that high DII scores were associated with higher risk of hip fracture in White women under 63 years of age [35]. Similarly, a prospective cohort study based on the Osteoarthritis Initiative data found that a pro-inflammatory diet (higher DII) is associated with higher fracture incidences among women but not among men [24]. A case-control study in China also found that a pro-inflammatory diet (a higher DII score) is associated with a higher risk of osteoporotic hip fracture. However, it reported significant results for both men and women, in contrast to the current study [36]. Also, whereas both the above-noted cohort studies and the case-control study had comparatively large population sizes, comparison of the case-control results with the present study is problematic, as a case-control study is more prone to recall bias. 

In addition to there just being far fewer men in the cohort overall, the rate of incident osteoporosis among men was much lower than women (i.e., among the 2572 individuals who developed osteoporosis, only 148 were men and 2424 were women). As the results of the present study are in such contrast to those of that earlier Chinese case-control study, further investigation is needed to clarify any association between the DII score and osteoporosis risk among men. Previous studies have suggested that due to the effect of sex hormones and genetic difference between men and women, female predominance in cases of autoimmune diseases such as osteoporosis is common [37,38]. Sex hormones alter the immune response, resulting in different disease phenotypes according to sex [39]. These sex differences in the immune response could be the reason for existing sex differences in the total number of osteoporosis cases. This would explain the overall sex difference in the rate of incident osteoporosis. However, the hazards ratios were nearly identical.

The present study also found even higher hazard ratios among pre- or peri-menopausal women as compared to post-menopausal women. However, because of the relatively small numbers of women at younger ages, the results were not statistically significant. While there are as yet no reports of any associations between the DII and osteoporosis among post-menopausal women, there are other reports that provide evidence for assuming both significant associations and their causes. Several researchers have found, for example, that estrogen deficiency increases bone resorption and also impairs bone formation [40,41]. During post-menopause, the ovaries have already reduced estrogen production, which can cause estrogen deficiency. Thus, a strong association between the DII and osteoporosis due to accelerated bone loss caused by estrogen deficiency in post-menopausal women can be inferred. However, after imputation in the present study, the association between the DII and osteoporosis risk among pre- or peri-menopausal women was not significant. Further research on the association between the DII and osteoporosis risk among pre- or peri-menopausal women is needed.

Whereas a statistically significant association between the DII and osteoporosis risk among participants who participated in irregular physical activity was found, no significant association among those with regular physical activity was evident in analyses, either of the total participants or of women. In contrast, men showed non-significant results, regardless of the regularity of their physical activity. Previous studies have uncovered a significant correlation between exercise and bone-mineral density, which is highly correlated with osteoporosis [42,43,44,45]. Researchers have suggested that muscle strength, which can be improved by exercise, is positively associated with BMD [42,43,44,45], especially among post-menopausal women. In the present study, however, a non-significant association was observed between the DII and osteoporosis among those with regular physical activity. One study suggested that exercise can play a beneficial role only when gonadal hormone levels are present [46]. Certainly, further investigation is needed in others, to clarify whether physical activity can influence the association between inflammatory diet and osteoporosis risk (low levels of BMD).

Another interesting finding is that calcium intake was higher among people with lower DII scores. This is consistent with the dogma in the West in that calcium prevents osteoporosis. However, it is important to understand which anti-inflammatory foods contribute to increased calcium intake in people with low DII scores [47,48].

Several studies [49,50,51,52] have shown that healthy dietary patterns (e.g., Mediterranean diet, DASH diet) are associated with bone health. A low DII (anti-inflammatory diet) score might be the result of healthy dietary patterns that satisfy anti-inflammatory parameters, such as antioxidant vitamins, and minerals, flavonoids, fiber, ginger, onion, garlic, pepper, tea, and others. [29]. It is conceivable that these components have an inflammatory-lowering effect, and in fact, a variety of nutraceuticals based on them have been shown to inhibit bone loss by several plausible mechanisms [53]. Most plant-derived components such as nutraceuticals and healthy dietary patterns (low DII) can provide, relative to a number of FDA-approved drugs, effective prevention, and therapy strategies, entailing few side effects.

The results of the present study should be considered in the light of its limitations. The principal shortcoming is that study participants were recruited at 38 health examination centers or training hospitals located in eight regions in Korea, and they tend to be more health conscious, which may have led to selection bias. Therefore it is difficult to generalize these results to the general Korean population. Second, the diagnosis of osteoporosis was self-reported by the participants, which may underestimate the incidence of osteoporosis with some errors. Although an FFQ is the most practical and common dietary assessment method used in prospective cohort studies, it contains a limited list of food items, and individuals are unable to accurately report their food intake retrospectively over a long period of time. Additionally, measurement error might have occurred in the usage of FFQ, and information on nine food parameters was not available for calculating the DII score, and it may influence the results. Furthermore, although the DII was designed to assess (estimate) the overall inflammatory potential of diet using a dietary assessment tool such as the FFQ, it is only a marker, not a direct biochemical inflammatory index such as hs-CRP, IL-1β, IL-4, IL-6, IL-10, or tumor necrosis factor (TNF)-α. Despite these limitations, the present study has strengths. It employed a prospective design, reducing the chance of recall bias, which yielded values that would be more representative of the entire population. It also had a large sample size, with a long-term follow-up, which yielded many incident cases of osteoporosis, especially in women. Also, it is a prospective cohort study, which reduced the chance of recall bias. Above all, the major strength of the present study is the fact that it is the first to have investigated, by means of the DII, the association between the inflammatory effects of diet with osteoporosis risk. In fact, it might have important clinical consequences, particularly given the trend toward more pro-inflammatory dietary patterns worldwide. By understanding and emphasizing the association between the DII and osteoporosis, healthier diets that can regulate inflammation, and thereby, benefit public health by lowering osteoporosis risk will be more effectively promoted. 

In conclusion, diets with higher inflammatory potentials were significantly associated with increased risk of osteoporosis in women, though not significantly in men. It is important to confirm these findings, especially given the lack of any similar previous study. Future studies of this type should be based on general public data and cohorts, including large numbers of men, as well as men and women living in rural areas. In the context of the continuing Westernization of diets in East Asian countries, the intake of less-inflammatory foods for prevention of osteoporosis is becoming more and more important.

## Figures and Tables

**Figure 1 nutrients-10-01999-f001:**
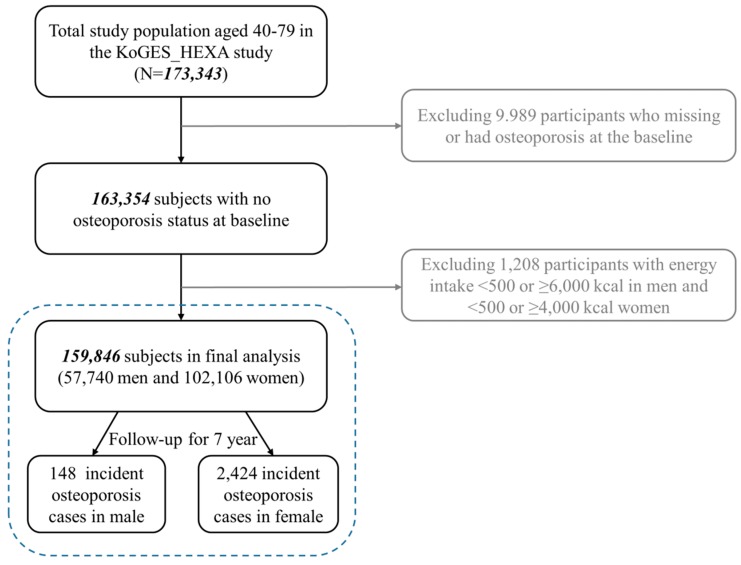
Flow chart of the analytical samples in the present study. KoGES_HEXA, Korean Genome and Epidemiology Study_Health Examinee study.

**Table 1 nutrients-10-01999-t001:** Baseline characteristics of participants by dietary inflammatory index (DII) quintiles in the KOGES cohort, 2001~2016.

Characteristics	Quintiles of Dietary Inflammatory Index (DII)					*p* Value ^d^
	Q1	Q2	Q3	Q4	Q5	
*N*	*n* = 31,894	*n* = 31,937	*n* = 31,971	*n* = 32,034	*n* = 32,000	
DII (Range)	−9.1296–−0.9826	−0.9824–0.3988	0.3989–1.2867	1.2868–2.1761	2.1762–7.1055	
Energy intake (Kcal/day)	2270.4(581.4) ^a^	1908.2(426.8)	1675.8(372.9)	1501.2(388.6)	1414.0(365.4)	<0.0001
Age at enrollment (year)	52.0(7.9)	52.1(8.1)	52.4(8.2)	53.0(8.4)	54.2(8.6)	<0.0001
Calcium intake (mg/day)	759.6(304.1)	506.4(170.4)	394.6(142.7)	311.7(144.8)	266.91(129.4)	<0.0001
Sex						
Men	11,070(34.7) ^b^	11,566(36.2)	11,636(36.4)	11,470(35.8)	11,998(37.5)	
Women	20,824(65.3)	20,381(63.8)	20,335(63.6)	20,564(64.2)	20,002(62.5)	<0.0001
BMI, kg/m^2 c^						
<18.5	559(1.8)	571(1.8)	600(1.9)	702(2.2)	901(2.8)	<0.0001
18.5–25	20,234(63.4)	20,641(64.6)	20,878(65.2)	20,989(65.6)	21,012(65.7)	
>25	11,104(34.8)	10,725(33.6)	10,530(32.9)	10,319(32.2)	10,081(31.5)	
Marriage						
Married	31,090(98.0)	31,114(97.9)	31,113(97.8)	31,109(97.6)	30,971(97.2)	<0.0001
Single/divorced	625(2.0)	674(2.1)	717(2.2)	782(2.4)	882(2.8)	
Education level						
~Elementary school	3588(11.4)	3988(12.7)	4838(15.3)	5831(18.5)	7912(25.1)	<0.0001
Middle~High school	18,633(59.2)	18,520(58.8)	18,341(58.1)	18,298(57.9)	17,694(56.1)	
College~	9247(29.4)	9003(28.5)	8397(26.6)	7477(23.6)	5925(18.8)	
Income (10,000 won)						
Less than 100	2031(7.8)	2169(8.0)	2660(9.9)	3406(12.6)	4615(16.8)	<0.0001
100~less than 200	4697(17.9)	4975(18.5)	5307(19.7)	5791(21.4)	6637(24.1)	
200~less than 300	6014(22.9)	6487(24.0)	6218(23.0)	6049(22.4)	5904(21.4)	
More than 300	13,478(51.4)	13,353(49.5)	12,790(47.4)	11,806(43.6)	10,380(37.7)	
Smoking status						
Never	23,198(73.1)	22,852(71.8)	22,902(71.9)	23,061(72.2)	22,005(69.0)	<0.0001
Past	4638(14.6)	5040(16.8)	5035(15.8)	4874(15.3)	4929(15.5)	
Current	3908(12.3)	3934(12.4)	3933(12.3)	3997(12.5)	4972(15.5)	
Alcohol consumption						
Never	15,406(48.5)	15,299(48.0)	15,578(48.9)	15,997(50.1)	16,405(51.4)	<0.0001
Past	1320(4.2)	1200(3.8)	1177(3.7)	1330(4.2)	1365(4.3)	
Current	15,042(47.3)	15,042(48.2)	15,135(47.4)	14,616(45.7)	14,141(44.3)	
Physical activity						
Irregular	12,476(39.2)	14,038(44.1)	14,881(46.7)	16,019(50.2)	18,249(57.2)	<0.0001
Regular ^e^	19,345(60.8)	17,828(55.9)	17,021(53.3)	15,910(49.8)	13,663(42.8)	
Menopausal status						
Post-	10,519(54.7)	10,558(54.6)	10,867(56.2)	11,535(58.9)	12,404(63.7)	<0.0001
Pre/peri-	8711(45.3)	8766(45.4)	8474(43.8)	8046(41.1)	7064(36.3)	

Quintile of the DII Score at baseline: Q1 indicates participants having the lowest dietary inflammatory index values, the least pro-inflammatory level; Q5 the highest, the most pro-inflammatory level. ^a^ The data are presented as means (standard deviation) for continuous variables. ^b^ The data were presented as n (%) for categorical variables. ^c^ BMI: body mass index. ^d^
*p* values for trends were calculated using the Jonckheere–Terpstra test for continuous variables, and the Mantel–Haenszel Chi-square test for categorical variables. ^e^ Regularity of physical activity was determined according to whether or not subjects participated regularly in any sports to the point of sweating.

**Table 2 nutrients-10-01999-t002:** Cox Proportional Hazard Ratios (HRs) (95% Confidence Intervals (CIs)) for osteoporosis risk by quintiles of DII score for all participants in the KOGES cohort, 2001~2016.

	Quintiles of Dietary Inflammatory Index (DII) ^a^					*P* Trend ^b^	*P* Int ^c^
	Q1	Q2	Q3	Q4	Q5		
**All subjects**							
Person-years	247,723	238,409	236,933	234,420	223,910		
Cases	440	491	517	578	546		
Crude HR (95% CI)	1.0	1.18 (1.03–1.34)	1.25 (1.10–1.42)	1.43 (1.26–1.61)	1.45 (1.28–1.64)	<0.0001	
Multivariate HR (95% CI) ^d^	1.0	1.21 (1.05–1.39)	1.25 (1.08–1.46)	1.38 (1.17–1.62)	1.33 (1.12–1.58)	0.0168	0.9136
All subjects with imputation							
Multivariate HR (95% CI) ^d^	1.0	1.19 (1.04–1.37)	1.25 (1.07–1.45)	1.37 (1.17–1.61)	1.32 (1.12–1.57)	0.0163	0.9955
**Men**							
Person-years	84,194	85,223	85,971	84,029	85,292		
Cases	22	30	25	37	34		
Crude HR (95% CI)	1.0	1.37 (0.79–2.37)	1.13 (0.64–2.00)	1.72 (1.02–2.92)	1.59 (0.93–2.72)	0.1237	
Multivariate HR (95% CI) ^d^	1.0	1.39 (0.76–2.54)	1.10 (0.56–2.15)	1.53 (0.77–3.04)	1.32 (0.64–2.71)	0.9183	
Men with imputation							
Multivariate HR (95% CI) ^d^	1.0	1.35 (0.74–2.44)	1.07 (0.55–2.07)	1.52 (0.78–2.99)	1.27 (0.63–2.60)	0.9795	
**Women**							
Person-years	163,529	153,186	150,962	150,391	138,618		
Cases	418	461	492	541	512		
Crude HR (95% CI)	1.0	1.20 (1.05–1.36)	1.30 (1.15–1.49)	1.45 (1.28–1.65)	1.53 (1.35–1.72)	<0.0001	
Multivariate HR (95% CI) ^d^	1.0	1.20 (1.04–1.39)	1.26 (1.08–1.48)	1.37 (1.16–1.62)	1.33 (1.11–1.59)	0.0147	
Women with imputation							
Multivariate HR (95% CI) ^d^	1.0	1.18 (1.03–1.37)	1.26 (1.08–1.47)	1.37 (1.16–1.61)	1.33 (1.11–1.58)	0.0136	

^a^ Quintile of the DII Score at baseline: Q1 indicates participants as having the lowest dietary inflammatory index values, the least pro-inflammatory level; Q5 the highest, the most pro-inflammatory level. ^b^ Continuous DII score was used to determine *p* for trend. ^c^
*p* value for interaction was calculated by contrasting the coefficients of the cross-product of menopausal status (pre/peri- and post-) and continuous DII score in the multivariable-adjusted time-dependent COX model. ^d^ Data are presented as hazard ratios (HRs) with correspondent 95% confidence intervals (CI). Multivariate-adjusted for sex (for all subjects); age (categorical); BMI (categorical); smoke (categorical); calcium intake (continuous); alcohol consumption (categorical); physical activity (categorical); energy intake (continuous).

**Table 3 nutrients-10-01999-t003:** Multivariate Cox Proportional Hazard Ratios (HRs) (95% Confidence Intervals (CIs)) for osteoporosis risk as stratified by menopausal status among women in the KOGES cohort, 2001~2016.

Menopausal Status	Quintiles of Dietary Inflammatory Index (DII) ^a^					P Trend ^b^	P Int ^c^
	Q1	Q2	Q3	Q4	Q5		
Pre/peri-menopause	1.0 ^d^	0.98 (0.67–1.43)	1.13 (0.73–1.77)	1.13 (0.73 -1.77)	1.39 (0.87–2.21)	0.1655	0.4033
Postmenopause	1.0 ^d^	1.24 (1.05–1.46)	1.36 (1.14–1.62)	1.41 (1.17–1.70)	1.33 (1.09–1.63)	0.0723	
Menopausal status with imputation							
Pre/peri-menopause	1.0 ^d^	0.97 (0.67–1.42)	0.98 (0.65–1.47)	1.14 (0.74–1.77)	1.37 (0.87–2.18)	0.1421	0.3522
Post-menopause	1.0 ^d^	1.23 (1.04–1.44)	1.35 (1.13–1.60)	1.40 (1.16–1.68)	1.33 (1.09–1.61)	0.0820	

^a^ Quintile of the DII Score at baseline: Q1 indicates participants having the lowest dietary inflammatory index values, the least pro-inflammatory level; Q5 the highest, the most pro-inflammatory level. ^b^ Continuous DII score was used to determine *p* for trend ^c^
*p* value for interaction was calculated by contrasting the coefficients of the cross-product of menopausal status (pre/peri- and post-) and continuous DII score in the multivariable-adjusted time-dependent COX model. ^d^ Data are presented as hazard ratios (HRs) with correspondent 95% confidence intervals (CI). Multivariate adjusted for age (categorical); BMI (categorical); smoke (categorical); calcium intake (continuous); alcohol consumption (categorical); physical activity (categorical); energy intake (continuous).

**Table 4 nutrients-10-01999-t004:** Multivariate Cox Proportional Hazard Ratios (HRs) (95% Confidence Intervals (CIs)) for osteoporosis risk as stratified by physical activity regularity in the KOGES cohort, 2001~2016.

Physical Activity	Quintiles of Dietary Inflammatory Index (DII) ^a^					P Trend ^b^	P Int ^c^
	Q1	Q2	Q3	Q4	Q5		
All subject							
Irregular	1.0 ^d^	1.26 (1.01–1.58)	1.39 (1.10–1.77)	1.47 (1.14–1.90)	1.49 (1.14–1.93)	0.0521	0.6866
Regular	1.0 ^d^	1.17 (0.98–1.41)	1.15 (0.94–1.41)	1.31 (1.05–1.63)	1.21 (0.96–1.53)	0.1408	
All subject with imputation							
Irregular	1.0 ^d^	1.27(1.02–1.59)	1.41 (1.12–1.79)	1.50 (1.17–1.93)	1.51 (1.16–1.95)	0.0345	0.5304
Regular	1.0 ^d^	1.16(0.97–1.39)	1.15 (0.94–1.41)	1.29 (1.04–1.60)	1.21 (0.96–1.52)	0.1749	
Men							
Irregular	1.0 ^d^	0.76(0.28–2.03)	0.83 (0.30–2.26)	0.75 (0.26–2.17)	0.86 (0.30–2.53)	0.6626	0.7958
Regular	1.0 ^d^	2.03(0.93–4.40)	1.33 (0.54–3.29)	2.53 (1.03–6.20)	1.67 (0.63–4.46)	0.5958	
Men with imputation							
Irregular	1.0 ^d^	0.72 (0.27–1.87)	0.80 (0.30–2.13)	0.73 (0.26–2.05)	0.84 (0.30–2.38)	0.5999	
Regular	1.0 ^d^	2.01 (0.93–4.37)	1.32 (0.54–3.27)	2.50 (1.02–6.13)	1.65 (0.62–4.40)	0.6155	0.6514
Women							
Irregular	1.0 ^d^	1.30 (1.03–1.64)	1.44 (1.12–1.84)	1.53 (1.18–1.99)	1.53 (1.17–2.01)	0.0340	
Regular	1.0 ^d^	1.13 (0.94–1.37)	1.15 (0.93–1.41)	1.26 (1.00–1.57)	1.19 (0.93–1.52)	0.1720	0.6398
Women with imputation							
Irregular	1.0 ^d^	1.32 (1.05–1.66)	1.47 (1.15–1.87)	1.57 (1.22–2.03)	1.56 (1.20–2.04)	0.0202	0.4769
Regular	1.0 ^d^	1.12 (0.93–1.34)	1.14 (0.93–1.40)	1.24 (0.99–1.55)	1.19 (0.93–1.51)	0.2093	

^a^ Quintile of the DII Score at baseline: Q1 indicates participants having the lowest dietary inflammatory index values, the least pro-inflammatory level; Q5 the highest, the most pro-inflammatory level. ^b^ Continuous DII score was used to determine p for trend. ^c^
*p* value for interaction was calculated by contrasting the coefficients of the cross-product of menopausal status (pre/peri- and post-) and continuous DII score in the multivariable-adjusted time-dependent COX model. ^d^ Data are presented as hazard ratios (HRs) with correspondent 95% confidence intervals (CI). Multivariate adjusted for age (categorical); BMI (categorical); smoke (categorical); calcium intake (continuous); alcohol consumption (categorical); physical activity (categorical); energy intake (continuous).

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
