# Peer review of "Positive Association between Dietary Inflammatory Index and the Risk of Osteoporosis: Results from the KoGES_Health Examinee (HEXA) Cohort Study"

_nutrients, 2018, doi:10.3390/nu10121999_

Round 1

Reviewer 1 Report

This study assessed the association between diets that are considered inflammatory and risk of osteoporosis. Justification needs to be given for the novelty of the study given that there are similar studies comparing this dietary pattern to fracture risk.

Line 65: change “accumulate” to “accumulates”

Lines 67-68: Change “…has been researched to verify the validity of the relationship between it and inflammatory markers such as CRP and IL-6” to “…has been validated against inflammatory markers such as CRP and IL-6”

Line 73: it is mentioned that the measure of dietary inflammation (i.e. DII) has been used to predict fractures…then on line 74 you indicate no studies have been done relating DII to osteoporosis. Fracture and osteoporosis are very closely related (i.e. osteoporosis essentially causes fracture); therefore I think you need to better highlight the novelty of the current study.

Line 108: it is stated the participants estimated food consumption over one year. Please clarify how often the food frequency questionnaire was administered. Was it administered once (to estimate the food intakes over the year) or was it administered several times throughout the year?

Line 178: Change “gender” to “sex”, as you use the latter term later in the paper.

Line 228-230: “Among women, those with irregular physical activity also had a significantly higher risk of osteoporosis (HR 1.53; 95% CI 1.17-2.01; Table 4), while those with regular physical activity showed insignificant results…” Are you referring to DII scores and their association with osteoporosis for women who did and did not participate in physical activity? Please clarify and re-write if this is the case. Also change “insignificant results” to “results that were not significant”

The same comment applies to lines 232-234.

Table 1: The range for DII scores overlaps for successive quintiles (for example, the upper range for Q1 is the same as the lower range for Q2).

Line 270-271: “The results confirmed the study’s hypothesis; i.e., that habitual dietary patterns influence inflammation.” I think this needs to be re-stated, as you did not measure inflammation in this study.

Lines 311-312: “However, after imputation in the present study, the association between the DII and osteoporosis risk among pre- or peri-menopausal women was significant.” Did you mean “not significant” in this sentence – see table 3?

Lines 315-317: “no significant association among those without regular physical activity was evident in analyses either of the total participants or of women.” Should “without regular physical activity” be changed to “with regular physical activity”?

Line 317: Change “Contrastingly” to “In contrast”; change “insignificant results” to “non-significant results”

Line 319: Change “which latter” to “the latter which”

Line 321: Change “insignificant association” to “the non-significant association”

Line 321-323: This sentence overall needs to be re-written for clarity.

Lines 324-325: “One study suggested that exercise can play a beneficial role only when gonadal hormone levels are present [46].” This is only one study. There are numerous studies that show that bone mineral density can be improved (or preserved) with exercise training in post-menopausal women.

Line 333: change “center” to “centers”

Line 334: change “hospital” to “hospitals”

Author Response

Response to reviewer 1

• This study assessed the association between diets that are considered inflammatory and risk of osteoporosis. Justification needs to be given for the novelty of the study given that there are similar studies comparing this dietary pattern to fracture risk.

Answer: Several epidemiologic studies have been conducted to examine the associations between the DII and various ailments such as fracture, cardiovascular diseases (CVD), and cancer [7, 21-24]. Osteoporosis is the main risk factor for fracture, and osteoporosis and fracture are closely related in clinical settings, nevertheless, osteoporosis is a potential target for efficient prevention of fracture. However, to the best of our knowledge, no studies on the possible associations between the DII and osteoporosis have yet been reported. It is unclear whether a more pro-inflammatory diet is associated with osteoporosis. Knowledge of the association of inflammatory diet with osteoporosis risk could be important to the tailoring of intervention strategies related to diet modulation for reduced inflammation. We corrected it in the revised (line 71-79).

• Line 65: change “accumulate” to “accumulates”

Answer: We corrected it in the revised (line 65).

• Lines 67-68: Change “…has been researched to verify the validity of the relationship between it and inflammatory markers such as CRP and IL-6” to “…has been validated against inflammatory markers such as CRP and IL-6”

Answer: We corrected it in the revised (line 67-68).

• Line 73: it is mentioned that the measure of dietary inflammation (i.e. DII) has been used to predict fractures…then on line 74 you indicate no studies have been done relating DII to osteoporosis. Fracture and osteoporosis are very closely related (i.e. osteoporosis essentially causes fracture); therefore I think you need to better highlight the novelty of the current study.

Answer: We corrected it in the revised (line 71-79).

• Line 108: it is stated the participants estimated food consumption over one year. Please clarify how often the food frequency questionnaire was administered. Was it administered once (to estimate the food intakes over the year) or was it administered several times throughout the year?

Answer: We corrected it in the revised (line 119).

• Line 178: Change “gender” to “sex”, as you use the latter term later in the paper

Answer: We corrected it in the revised (line 188), in table 2 legends.

• Line 228-230: “Among women, those with irregular physical activity also had a significantly higher risk of osteoporosis (HR 1.53; 95% CI 1.17-2.01; Table 4), while those with regular physical activity showed insignificant results…” Are you referring to DII scores and their association with osteoporosis for women who did and did not participate in physical activity? Please clarify and re-write if this is the case. Also change “insignificant results” to “results that were not significant”

Answer: We corrected it in the revised (line 238-240). Women did not participate in regular physical activity had a significantly higher risk of osteoporosis (HR 1.53; 95% CI 1.17-2.01; Table 4), while women with regular physical activity showed not significant results (HR 1.19; 95% CI 0.93-1.52; Table 4).

• The same comment applies to lines 232-234.

Answer: We corrected it in the revised (line 242-243).

• Table 1: The range for DII scores overlaps for successive quintiles (for example, the upper range for Q1 is the same as the lower range for Q2).

Answer: We corrected it in the revised Table 1.

• Line 270-271: “The results confirmed the study’s hypothesis; i.e., that habitual dietary patterns influence inflammation.” I think this needs to be re-stated, as you did not measure inflammation in this study.

Answer: We corrected it in the revised (line 276-277).

 • Lines 311-312: “However, after imputation in the present study, the association between the DII and osteoporosis risk among pre- or peri-menopausal women was significant.” Did you mean “not significant” in this sentence – see table 3?

Answer: We corrected it in the revised (line 322).

• Lines 315-317: “no significant association among those without regular physical activity was evident in analyses either of the total participants or of women.” Should “without regular physical activity” be changed to “with regular physical activity”?

Answer: We corrected it in the revised (line 327).

• Line 317: Change “Contrastingly” to “In contrast”; change “insignificant results” to “non-significant results”

Answer: We corrected it in the revised (line 328).

• Line 319: Change “which latter” to “the latter which”

Answer: We corrected it in the revised to “which” (line 330).

• Line 321: Change “insignificant association” to “the non-significant association”

Answer: We corrected it in the revised (line 333).

• Line 321-323: This sentence overall needs to be re-written for clarity.

Answer: We corrected it in the revised (line 329-337).

• Lines 324-325: “One study suggested that exercise can play a beneficial role only when gonadal hormone levels are present [46].” This is only one study. There are numerous studies that show that bone mineral density can be improved (or preserved) with exercise training in post-menopausal women.

Answer: based on the reviewer’s comment, we added some discussion of other study (329-337).

• Line 333: change “center” to “centers”

Answer: We corrected it in the revised (line 352)

• Line 334: change “hospital” to “hospitals”

Answer: We corrected it in the revised (line 353)

Reviewer 2 Report

To:

Editorial Board

Nutrients

Title: “Positive association between dietary inflammatory index and risk of osteoporosis: Results from the KoGES_Health Examinee study (HEXA) cohort study”.

Dear Editor,

I read this manuscript and I think that:

-          A flow chart of the study should be added to the paper.

-          Inclusion and exclusion criteria should be better specified.

-          The use of questionnaires is a limitation of the study design. This should be discussed in a dedicated limitation section. Please provide.

-          The role of nutraceuticals on health should be discussed. The authors can consider the paper from Scicchitano P et al. Journal of Functional Foods 2014;6:11-32.

Author Response

Response to reviewer 2

• A flow chart of the study should be added to the paper.

Answer: We added flow chart in the revised (line 108, 112-115 in Figure 1)

• Inclusion and exclusion criteria should be better specified.

Answer: We corrected it in the revised (line 91-92, 101-108)

• The use of questionnaires is a limitation of the study design. This should be discussed in a dedicated limitation section. Please provide.

Answer: Although the DII was designed to assess (estimate) the overall inflammatory potential of diet using a dietary assessment tool such as the FFQ, it is only a marker, not a direct biochemical inflammatory index such as hs-CRP, IL-1β, IL-4, IL-6, IL-10 or tumor necrosis factor (TNF)-α. We added it in the revised (line 361-364)

• The role of nutraceuticals on health should be discussed. The authors can consider the paper from Scicchitano P et al. Journal of Functional Foods 2014;6:11-32.

Answer: Several studies [49-52] have shown that healthy dietary patterns (e.g., Mediterranean diet, DASH diet) are associated with bone health. A low DII (anti-inflammatory diet) score might be the result of healthy dietary patterns that satisfy anti-inflammatory parameters such as antioxidant vitamins, and minerals, flavonoids, fiber, ginger, onion, garlic, pepper, tea and others. [29]. It is conceivable that these components have a reduced inflammatory power, and in fact, a variety of nutraceuticals based on them have been shown to inhibit bone loss by several plausible mechanisms [53]. Most plant-derived component such as nutraceuticals and healthy dietary patterns (low DII) can provide, relative to a number of FDA-approved drugs, effective prevention and therapy strategies entailing few side effects. We added it in the revised (line 342-350)

Round 2

Reviewer 1 Report

The authors have adequately addressed most of the previous comments. A few minor revisions are recommended:

Lines 240 and 243: Change “insignificant” to “non-significant”

In Table 1, there is still overlap for the DII range between Quintile 1 (-9.1296 – -0.9826) and Quintile 2 (-0.9824 – 0.3988)

Line 344: Change “a reduced inflammatory power” to “an inflammatory-lowering effect”

Author Response

Response to reviewer 1

Lines 240 and 243: Change “insignificant” to “non-significant”

Answer: We corrected it in the revised (line 240, line 243).

In Table 1, there is still overlap for the DII range between Quintile 1 (-9.1296 – -0.9826) and Quintile 2 (-0.9824 – 0.3988)

 Answer: We think DII range between Quintile 1 (-9.1296 – -0.9826) and Quintile 2 (-0.9824 – 0.3988) was not overlapped.

Line 344: Change “a reduced inflammatory power” to “an inflammatory-lowering effect”

Answer: We corrected it in the revised (line 346).

For English editing, we’ve our manuscript checked by a professional English editing service.

Reviewer 2 Report

To:

Editorial Board

Nutrients

Title: “Positive association between dietary inflammatory index and risk of osteoporosis: Results from the KoGES_Health Examinee (HEXA) cohort study”

Dear Editor,

I read the revised version of this manuscript and I think that the authors well addressed my previous comments. The paper improved very much.

Author Response

Thank you so much.